# Ocular Sporotrichosis

**DOI:** 10.3390/jof7110951

**Published:** 2021-11-10

**Authors:** Max Carlos Ramírez-Soto, Andrés Tirado-Sánchez, Alexandro Bonifaz

**Affiliations:** 1School of Public Health and Administration, Universidad Peruana Cayetano Heredia, Lima 15102, Peru; 2Facultad de Ciencias de la Salud, Universidad Tecnológica del Perú, Lima 15046, Peru; 3Internal Medicine Department, Hospital General de Zona 29, Instituto Mexicano del Seguro Social, Ciudad de México 07950, Mexico; atsdermahgm@gmail.com; 4Dermatology Service & Mycology Department, Hospital General de México, “Dr. Eduardo Liceaga”, Balmis 148, Colonia Doctores, Ciudad de México 06726, Mexico; a_bonifaz@yahoo.com.mx

**Keywords:** sporotrichosis, *Sporothrix*, ocular infection, conjunctivitis, ocular adnexa, endophthalmitis

## Abstract

Sporotrichosis is a subacute or chronic mycosis predominant in tropical and subtropical regions. It is an infection of subcutaneous tissue caused by *Sporothrix* fungus species, but occasionally resulting in an extracutaneous condition, including osteoarticular, pulmonary, nervous central system, and ocular disease. Cases of ocular sporotrichosis are rare, but reports have been increasing in recent decades. Ocular infections usually occur in hyperendemic areas of sporotrichosis. For its classification, anatomic criteria are used. The clinical presentation is the infection in the ocular adnexal and intraocular infection. Ocular adnexa infections include palpebral, conjunctivitis, and infections of the lacrimal sac. Intraocular infection includes exogenous or endogenous endophthalmitis. Most infections in the ocular adnexal have been reported in Brazil, China and Peru, and intraocular infections are limited to the USA and Brazil. Diagnosis is performed from *Sporothrix* isolation in the mycological examination from ocular or skin samples. Both sporotrichosis in the ocular adnexa and intraocular infection can mimic several infectious and non-infectious medical conditions. Ocular adnexa infections are treated with potassium iodide and itraconazole. The intraocular infection is treated with amphotericin B. This review describes the clinical findings and epidemiological, diagnosis, and treatment of ocular sporotrichosis.

## 1. Introduction

Sporotrichosis is a subacute or chronic infection predominant in the subcutaneous tissue [1]. It is caused by *Sporothrix* species fungal and is predominant in tropical and subtropical regions [2,3]. Most cases are reported from Brazil, China, Peru, and Mexico [2,4]. The upper limbs, lower limbs, and the face are affected [1,3]. Occasionally, it can also result in an extracutaneous infection, including osteoarticular, pulmonary, nervous central system, and ocular infection [4,5,6,7]. Ocular infection due to *Sporothrix* manifests with a lesion in the ocular adnexa or the deep eyeball structures, causing endophthalmitis [8,9]. Cases of ocular sporotrichosis are rare, but reports have been increasing in recent decades [8,9,10,11,12]. Intraocular infection due to *Sporothrix* could be a sight-threatening condition that may result in vision loss. To date, little is known about the epidemiology, clinical findings, and outcomes of ocular sporotrichosis. The purpose of this review is to discuss the epidemiology, the aspects related to etiology, main clinical manifestations, diagnosis, and management of ocular sporotrichosis.

## 2. Epidemiology and Etiology

Sporotrichosis is a significant health problem in tropical and subtropical regions. Sporotrichosis has a wide geographic distribution worldwide, but most cases have been reported in retrospective studies from Brazil, China, Mexico, and Peru (Figure 1) [2,4,13,14,15,16]. Sporotrichosis has been described in humans and animals like cats in Brazil, the United States, Malaysia, and Argentina. Isolated cases in cats also have been reported in Germany, Japan, Mexico, Spain, India, and Australia [17,18,19,20]. Brazil seems to be the most endemic country for sporotrichosis in the world. More than 5000 human cases and 5113 feline cases have been reported in Brazil from 1998 to 2018 [13]. In retrospective studies from China, 2000 cases were recorded in 9 years [14], and more than 1800 cases in Peru [21]. Ocular infections due to *Sporothrix* occur in hyperendemic areas. Most infections in the eyelid have been reported in China and Peru [8,11], and conjunctival sporotrichosis in Brazil, a hyperendemic area of zoonotic transmission [12]. Intraocular infections are limited to the USA and Brazil (Figure 1) [9]. Ocular sporotrichosis affects all age groups and both sexes [8,9,11,12]. Although *Sporothrix* can enter the skin or eye via a traumatic inoculation with vegetal material or a wood splinter, or contact with cats or cats with sporotrichosis, most cases occur in the absence of predisposing factors.

Infections are usually caused by *S. schenckii*, *S. globosa*, and *S. brasiliensis*. Rare pathogens include *S. pallida* and *S. mexicana*. *S. globosa* causes about 99.3% of cases in Asia, 94% in Australia and South Africa by *S. schenckii*, 88% in Brazil by *S. brasiliensis*, and 89% in South and Central America and North America by *S. schenckii* [2,3]. All these species grow in the soil at temperatures of 6.6–28.84 °C and 37.5–99.06% humidity, and associated with a variety of plants, flowers, decaying material, woody debris, reed leaves, corn stalks, leaves, and wood crumbs, potentially facilitating its establishment and proliferation in the environment [22]. *S. schenckii* and *S. brasiliensis* constitute the causal agents of ocular sporotrichosis [8,9,11,12], but the causal agent has not been typified in most cases. Additionally, a case of fungal keratitis was caused by *S. pallida*, a rare human pathogen [23].

## 3. Clinical Presentation

Classically, cutaneous sporotrichosis presents with lymphocutaneous, fixed, and disseminated lesions. Nearly 80% of the affected patients show the lymphocutaneous form [1,6]. After traumatic implantation, the lesion may ulcerate and progress along the regional lymphangitic channels. Later, these nodules may ulcerate. The fixed cutaneous sporotrichosis is a single ulcer with no regional lymphatic spreading [1,6]. Disseminated or hematogenous sporotrichosis is rare and usually seen in immunocompromised patients as an opportunist infection. Disseminated sporotrichosis can extend to various organs and systems, including the central nervous system, osteoarticular tissue, pulmonary, and ocular, progressing to fungemia [5,24]. Extracutaneous sporotrichosis is rare and often develops ocular, pulmonary, and central nervous system involvement (Figure 2) [5,7].

For the classification of ocular sporotrichosis, anatomic criteria and the source of infection are used. The clinical presentation of ocular sporotrichosis is ocular adnexal infection and intraocular infection (Figure 2). Ocular adnexa infections include palpebral, conjunctivitis, and disorders of the lacrimal sac (Figure 3A–H). The intraocular disease consists of exogenous or endogenous endophthalmitis. Most ocular conditions include skin lesions, except for primary conjunctivitis, dacryocystitis, and exogenous endophthalmitis (Figure 2). The site of infection, the factors as traumatic inoculation, the residence in the hyperendemic area and host factors that include immune responses determines the clinical outcome of infection (Table 1).

### 3.1. Ocular Adnexa Sporotrichosis

The term ocular adnexa includes the tissues and structures surrounding the eye, including the orbital soft tissue, lacrimal system, conjunctiva, eyelids, and eyebrows [25]. Sporotrichosis in the ocular adnexa consists of a group of infections that can be classified into (1) palpebral infections, one group of infections of the dermis around the eyes, (2) conjunctivitis, one group of infections of the bulbar and palpebral conjunctiva, and (3) infections of the lacrimal system. The eyelid and conjunctiva are more affected in sporotrichosis in the ocular adnexa [8,12], followed by a lacrimal sac that is less affected [8]. Usually, dermatologists manage palpebral lesions and infections of the conjunctiva and lacrimal system by ophthalmologists with oculoplastic expertise.

#### 3.1.1. Sporotrichosis Palpebral

Eyelid sporotrichosis is caused by *S. schenckii* and also by *S. brasiliensis* and *S. globosa* (Table 2) [8,11,26,27,28,29,30,31,32,33,34,35,36]. Among the adnexa ocular, 82% of these cases are limited to the eyelids [8]. A case series reported the epidemiologic characteristics and clinical features of 72 patients with eyelid sporotrichosis in Jilin, China. This study included 43 children and 29 adults between 2 months and 80 years. Fixed cutaneous lesion occurred in 57%, lymphocutaneous in 35%, and disseminated in 8% of cases. The 43% of patients had a history of trauma caused by vegetal material and wood [11]. A systematic review included 19 patients with eyelid sporotrichosis and 2 in eyebrows in a hyperendemic area in Peru. In this study, 57.1% of patients were male, and 87.5% were between 0 and 14 years. Lymphocutaneous lesion occurred in 62% and fixed form in 38% of cases [8]. A series of 16 cases of eyelid sporotrichosis has also been reported in another Peruvian hyperendemic area (La Libertad). In this series, most of the cases were lymphocutaneous [29]. Other case series in China reported 10 cases of eyelid sporotrichosis, lymphocutaneous (6 cases), fixed cutaneous (3 cases), and eyelid abscess (1 case) [28]. In Brazil, case reports of sporotrichosis in eyelids are increasing [10,26]. Isolated cases also have been reported in Mexico, Costa Rica, Japan, Argentina, Malaysia, and Australia (Table 2) [30,31,32,33,34,35]. The eyelid lesions may be primary, or their involvement may be part of lymphocutaneous or disseminated lesions. These lesions start as a small subcutaneous nodule at the site of an inoculation that later is ulcerated or spreads along the regional lymphangitic channels.

Skin eyelid lesions may be papular, nodular, ulcerative, or infiltrative (plaque-like) or may show a combination of these features (Table 2) [8]. However, not all eyelid lesions present with these characteristic clinical manifestations. In China, some eyelid lesions were granuloma annulare-like plaque, abscess, and cyst-like lesions [12,28].

In patients with disseminated infection, eyelid lesions were manifested as a cluster of papules and verrucous plaques [12,28]. The eyelid is the first line of defense for the eye. Therefore, palpable lymphadenopathy or an ulcerated lesion in the periocular region in pediatric patients or some adults from endemic and hyperendemic areas are a suggestive finding of palpebral sporotrichosis. These epidemiological and clinical features may prompt clinicians to consider sporotrichosis in eyelids and facilitate the diagnosis, management, and differentiation of other periocular infections.

#### 3.1.2. Conjunctival Sporotrichosis

Twenty-one studies (case reports and case series) reported ocular sporotrichosis with involvement of the conjunctiva, representing a total of 56 patients, with a mean age of 33.33 ± 21.81 years (range 3–80 years) (Table 3) [34,37,38,39,40,41,42,43,44,45,46,47,48,49,50,51,52,53,54,55]. Most cases have been reported in Brazil (44 cases) and Malaysia (8 cases). Isolated cases also have been reported in the USA, Japan, Thailand, and Colombia. Among them, 70% were women (39 patients). The left eye was more commonly affected than the right one (15 vs. 12 cases, respectively), although in 29 cases, the affected eye was not specified. *S. schenckii* was isolated in 25% of the cases (14 patients) [39,40,41,44,47,50,51,52,53] followed by *S. brasiliensis* (5.3%, 3 cases) [26,45]; isolation was not achieved in 4 patients (7.4%) [3,4,7,11], and in 35 (62.5%), the species involved was not determined [12,34,37,38,43,49]. Previous trauma related to conjunctival implantation was not reported in any patient, although contact with cats was reported in 50 patients (89.3%) (Table 3) [26,34,37,38,40,41,43,44,45,46,47,48,49,50,51,52,53].

The tarsal conjunctiva was more frequently affected than the bulbar (49 cases, 87.5% versus six patients, 10.7%, respectively) (Table 4). Eyelid involvement was reported in 18 cases (33.3%), the lower eyelid being more affected than the upper one (18 cases (32.1%) versus four patients (7.1%), respectively). In 50% of the cases (28 subjects), a Parinaud oculoglandular syndrome was integrated. The most frequently reported lymph node involvement was preauricular lymphadenopathy (14 cases, 25%), although the affected lymph nodes were not specified in 18 patients (32.1%) (Table 3). The response to treatment was favorable in cases of palpebral and bulbar disease (79% and 75%, respectively), although the doses of itraconazole used show great variability according to the severity of the case and the low number of patients and clinical studies, it does not allow for more informed conclusions (Table 4).

#### 3.1.3. Dacryocystitis Due to *Sporothrix*

Among the ocular adnexa, infection of the lacrimal sac due to *Sporothrix* is infrequent. To date, only six cases have been identified in the published literature [56,57,58]. There are five cases in Brazil, a hyperendemic area of sporotrichosis associated with zoonotic transmission [56,57], and one in Jilin, China [58]. The age of the cases ranged from 2 to 41 years, and five were women. Both the right and left eyes were affected similarly, and three cases presented compromise of the conjunctiva [56,58]. *Sporothrix* spp., the causal agent, has been identified in four patients, but *S. schenckii* (1 case) and *S. brasiliensis* (1 case) have been identified in culture. Among these cases, no trauma associated with ocular implantation has been identified [56,57,58]. Exceptionally, in one case, contact lens use was reported [58], and in other cases, contact with cats [56,57] with no specific history of injury.

### 3.2. Intraocular Sporotrichosis

Intraocular infection is exceedingly rare, with only isolated case reports and a systematic review in the published literature [9,59,60,61,62,63,64,65,66,67,68,69,70,71,72,73,74]. Intraocular sporotrichosis can present as endophthalmitis, granulomatous uveitis, scleritis, retinitis, choroiditis, or iridocyclitis [59,60,61,62,63,64,65,66,67,68,69,70,71,72,73,74]. According to the source of infection, intraocular sporotrichosis may be either exogenous or endogenous. In exogenous endophthalmitis, *Sporothrix* fungus is introduced into the eye via trauma with vegetable matter or from an external source. Endogenous endophthalmitis occurs due to hematogenous seeding of *Sporothrix* fungus in the eye or systemic infection or disseminated sporotrichosis [9].

#### 3.2.1. Exogenous Endophthalmitis

Exogenous endophthalmitis due to *Sporothrix* is less frequent than endogenous endophthalmitis [9]. Exogenous endophthalmitis occurs after eye trauma penetrating, although most exogenous cases do not report penetrating ocular trauma [59,60,61,62,63,64,65]. All patients have been reported in the USA, and *Sporothrix* spp. is the most common etiological agent [9,59,60,61,62,63,64,65]. Risk factors include delay in treatment and the presence of a lacerating injury resulting from the ocular trauma [60]. The clinical presentation may be subacute or chronic. Anterior uveitis is a clinical manifestation that is more common in exogenous endophthalmitis, including granulomatous uveitis and scleritis [59,60,61,62,63,64,65]. Clinical findings can be nonspecific, emulating non-infectious forms of uveitis. Usually, exogenous endophthalmitis results from prolonged chronic fungal infection/uncontrolled, since most case reports are diagnosed within 100 days of symptom onset (Table 5). Patients often present with decreased vision and redness, and some also have eye pain, leading to a delay in diagnosis and treatment [62,63,64,65]. An eye examination usually reveals a hypopyon and intraocular inflammation [63,64,65]. Exogenous endophthalmitis is a severe eye infection. These cases are medical emergencies, as delay in treatment may result in permanent vision loss.

#### 3.2.2. Endogenous Endophthalmitis

Endogenous endophthalmitis due to *Sporothrix* is a severe but uncommon cause of intraocular inflammation [66,67,68,69,70,71,72,73,74]. This disease is caused by hematogenous dissemination of the *Sporothrix* fungus spread of cutaneous or disseminated sporotrichosis to the eye. *S. schenckii* is the most common cause of endogenous endophthalmitis [66,69,70,71,72,73,74], although there have been cases reported of endogenous endophthalmitis caused by *S. brasiliensis*, an emergent pathogen associated with the zoonotic transmission of sporotrichosis in Brazil [67,68]. The clinical findings include choroiditis, chorioretinitis, uveitis, and retinitis [67,70,73,74]. A systematic review reveals that posterior uveitis seems to be a clinical manifestation more common in endogenous endophthalmitis caused by *Sporothrix*, especially in HIV-infected patients from hyperendemic areas (Table 5) [9]. The initial manifestation is usually posterior uveitis since it is highly vascular, and as a consequence, the intraocular infection usually starts in the posterior segment. Multifocal choroiditis due to *S. brasiliensis* was also reported in five eyes of three immunocompromised patients with disseminated sporotrichosis in Brazil [68]. In contrast to patients with exogenous endophthalmitis, all patients with endogenous endophthalmitis have an identifiable systemic infection. Systemic infections include sporotrichosis disseminated, osteoarticular, and widespread multiorgan dissemination (including cutaneous, osteoarticular, and pulmonary) [66,67,68,69,70,71,72,73,74]. Patients with endogenous endophthalmitis may present with varying inflammation, decreased vision, and visual loss in some cases. In some cases, diagnosis delay can worsen ocular outcomes, as the disease can disseminate anterior or posterior uveitis to endogenous endophthalmitis [74].

## 4. Differential Diagnoses

Ocular sporotrichosis can mimic a broad spectrum of diseases that are infectious and non-infectious. In endemic and hyperendemic area, it is vital to differentiate ocular sporotrichosis from other endemic diseases. Diagnosis must be confirmed by mycological examination. The differential diagnoses for sporotrichosis in ocular adnexa include palpebral lesions caused by the fungus and conjunctivitis caused by bacterial or viral infections. Intraocular sporotrichosis should be included in the differential diagnosis for ocular inflammation, endophthalmitis, choroiditis, retinitis, and uveitis caused by bacterial and fungal infections or viral (Table 6).

## 5. Laboratory Diagnosis

The gold standard for diagnosis of ocular sporotrichosis includes the isolation and identification of the etiological agent. Biological material is usually obtained from exudate in lesions, scales, tissue fragments, a biopsy of the cutaneous lesion in palpebral lesions, or swabs from the conjunctival mucosa [8,11,12]. Biological material in intraocular infections includes vitreous and aqueous humor samples, scleral and corneal scraping, and purulent eye discharge [9,59,60,61,62,63,64,65,66,67,68,69]. The usual Sabouraud dextrose agar with or without antibiotics, incubated at 28 °C, is the most used; colonies appear in an average time of five to eight days. *Sporothrix* sp. colonies develop in three to five days, being limited, membranous, radiated and whitish, beige, or pigmented (but not included in the black fungi group) [1,5,75]. Species most commonly isolated in the ocular infections are *S. schenckii, Sporothrix* spp. [8,11,12,59,60,61,62,63,64,65,66,67,68,69], *S. brasiliensis* in Brazil [26,67,68], and *S. globosa* in China [27].

In the patient samples, the observation of *Sporothirx* spp. yeasts are made with direct or fresh examination with conventional stains including Giemsa, PAS, and Grocott (KOH (10%) or NaOH (4%) is rarely used due to poor results); the observation of yeast in cigar-shaped buds or small boats is characteristic, but infrequently seen (5–10%), except in disseminated cases and immunosuppressed patients. Cat and dog cases have many yeasts; however, this is frequent only in the ocular conjunctiva samples in human subjects [1,5,10,12,47].

Like direct examination, biopsy (skin) is not pathognomonic; on rare occasions, asteroid bodies with yeasts may appear. In general, the histopathological image is a combination of suppurative granulomatous conformation and pyogenic reaction, and to a lesser extent, a tuberculoid granuloma can be observed. Large numbers of yeast can only be observed in disseminated cases that affect the eye and adjacent structures [5,75].

Other supporting tests are the intradermal reaction (IDR) with sporotrichin M (mycelial); it should be noted that it is not a standardized test (performed in-house), and in many countries, it is not authorized. It is carried out with the polysaccharide metabolic fraction of *S. schenckii* (peptide-rhamnomannan); it is applied intradermal (dilution 1:2000, averaging 5 × 10^7^ cells/mL) [5,75,76]. The reading is done with the same criteria as the PPD. This test is immensely guiding in above 90%. Serology has little value, more useful in disseminated cases; the most used techniques are precipitins, agglutinins, and complement fixation [1,5].

Molecular biology is of great importance in the identification of strains from tissue samples (biopsies) through the amplification of DNA fragments, with PCR tests (chitin synthetase gene, ChS1, 26S rDNA gene, and topoisomerase II gene), and PCR-RFLP (polymerase chain reaction-restriction fragment length polymorphism) [1,75]. Proteomic identification, particularly with MALDI-TOF MS techniques, makes it possible to effectively and rapidly identify the five species of *Sporothrix* [1,75].

## 6. Treatment

Like cutaneous sporotrichosis, the primary ocular condition or associated by extension of cutaneous or disseminated cases is managed as follows: Itraconazole is the drug considered of choice by the North American Guidelines for the Treatment of Sporotrichosis, especially for lymphangitic and disseminated cases, and it is regarded as the first option in ocular disease cases [77]. It is used at doses of 100–300 mg/day, depending on the weight and conditions of the patient; the treatment time fluctuates between four and six months, with minimal collateral effects, and it is essential to mention that its absorption depends on gastric pH and food [5,77].

Another highly active drug is potassium iodide (KI). It is the therapy of choice in underdeveloped countries due to its excellent efficacy, low side effects, easy administration, and low cost [1,5,78]. It is generally managed in a saturated solution of KI and drops starting with 2–20 three times a day; in the various series, tanning is obtained within 2–3 months of treatment; there are some cases with rapid responses (15 days). In general, the most critical side effect is gastritis, and to a lesser degree, rhinitis, bronchitis, urticaria, and erythema nodosum. It should be mentioned that KI is prepared as a magisterial formula. In disseminated cases and immunosuppressed patients, they do not respond [78,79,80]. In ocular cases, it also has been administered in combination with KI and itraconazole, especially in those that did not respond well to monotherapy [81].

Amphotericin B is the treatment of choice for systemic, disseminated sporotrichosis cases and must be administered in the hospital. Management with liposomal amphotericin B is suggested, at a standard dose of 3 mg/kg/day, with a range of 3–5 mg/kg/day. For amphotericin B deoxycholate the dose is 0.25–0.75 mg/kg/day. Treatment time is variable and depends more on side effects (kidney) [1,5,77]. Systemic amphotericin B alone or in combination with an oral antifungal is the treatment most commonly used in intraocular infections. It can also be used the intravitreal and systemic antifungal or amphotericin B. It is important to note that for the management of amphotericin B, the patient must be hospitalized, and remember that the most important collateral effects are phlebitis and especially kidney damage; for topical management, it is usually an irritating drug in the mucous membranes [9,77]. The subconjunctival and oral agents show poor intraocular penetration; therefore, intravitreal and systemic antifungal therapy should be considered in cases of endophthalmitis. Cases of exogenous endophthalmitis have worse ocular outcomes and complications compared with endogenous endophthalmitis. The endophthalmitis complications include blindness and irreversible vision loss [9,60,61,62,63,64].

Another of the drugs used is terbinafine, at doses of 250–500 mg/day; the responses are variable, and it is considered a less active drug than itraconazole but with fewer drug interactions and side effects [1,5]. There are varying results with the administration of fluconazole; its dose is 200–400 mg/day [1]. The use of corticosteroids in cases surrounding the eye is beneficial to avoid the appearance of keloid and fibrous lesions, and they should always be associated with systemic antifungals. The most widely used is prednisone at doses of 10–25 mg/day [5].

## 7. Conclusions

Sporotrichosis is an emerging mycosis subcutaneous around the world, which occasionally can result in an ocular infection. Traumatic inoculation with vegetable material, contact with cats, HIV infection, residence in hyperendemic areas, and disseminated infection are associated with this disease. Ocular lesions must be classified using anatomic criteria and the source of infection. There are two defined types of lesions: ocular adnexal lesions, including palpebral lesions, conjunctivitis, and dacryocystitis, and intraocular infection that include endophthalmitis, uveitis, and choroiditis. The diagnosis is based on suggestive ocular findings and mycological examination from ocular or skin samples. Itraconazole and KI are the most used antifungal agents in treating sporotrichosis in adnexa ocular, and amphotericin B is the antifungal agent for treating intraocular infection. Ocular adnexal lesions have an excellent clinical outcome, whereas intraocular infections may have worse results and complications. This review has identified some key points to improve ocular sporotrichosis′s clinical, epidemiological, and therapeutic aspects. These findings have public health implications and can assist healthcare providers involved in eye care.

## Figures and Tables

**Figure 1 jof-07-00951-f001:**
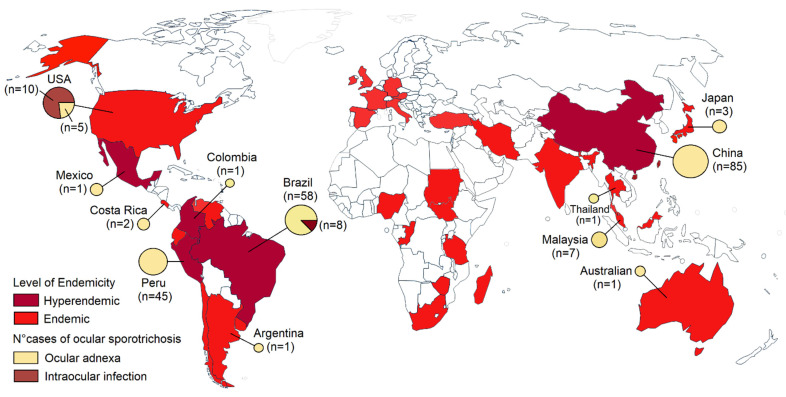
Map of the global distribution of sporotrichosis and cases of ocular sporotrichosis published in the literature.

**Figure 2 jof-07-00951-f002:**
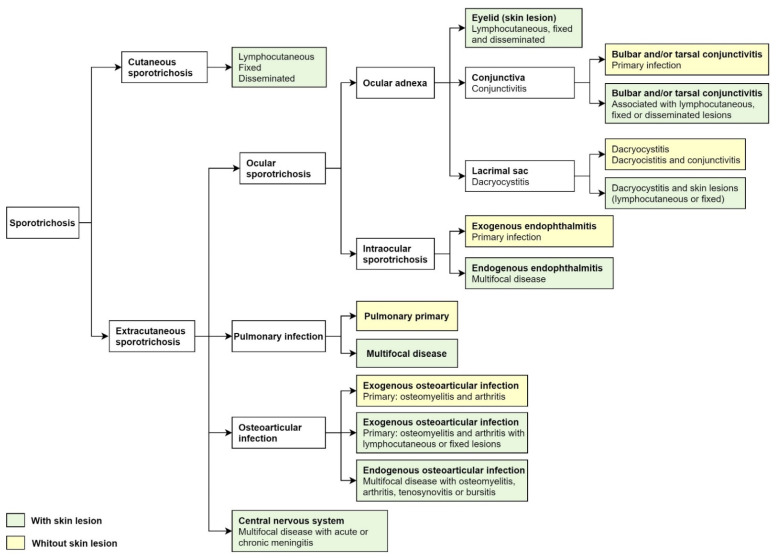
Cutaneous and extracutaneous sporotrichosis. For the classification of ocular sporotrichosis, anatomic criteria and source infection.

**Figure 3 jof-07-00951-f003:**
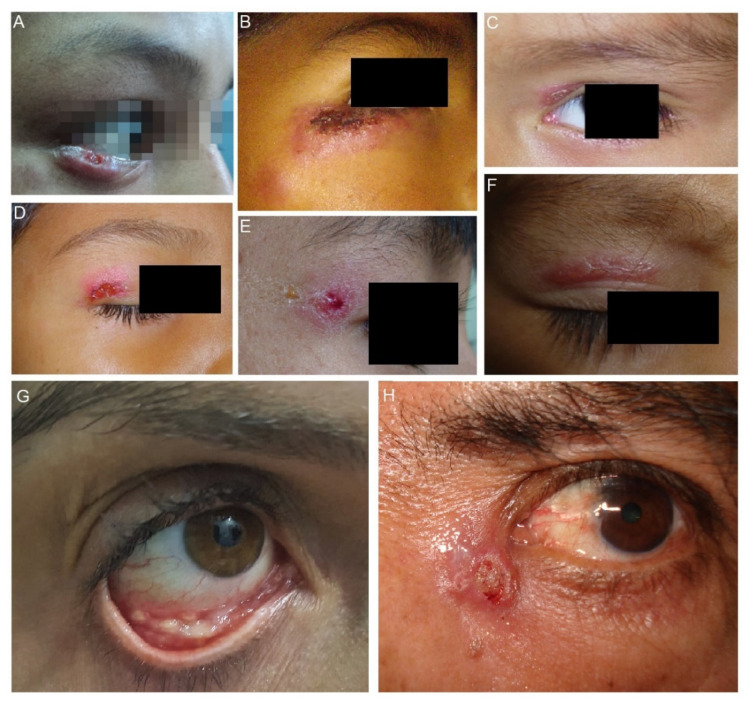
Clinical types of lesions observed in patients with ocular sporotrichosis. (**A**,**B**) Lymphocutaneous eyelid sporotrichosis is characterized by the emergence of an indurated papule and nodules contiguous to the initial lesion. (**C**–**E**) Fixed sporotrichosis in upper eyelid. (**F**) Fixed sporotrichosis; granuloma-like plaque in upper eyelid. (**G**) Granulomas in the lower right palpebral conjunctiva (Courtesy of Isabella D. Ferreira-Gremião, Fiocruz, Rio de Janeiro, Brazil). (**H**) Dacryocystitis in the left eye.

**Table 1 jof-07-00951-t001:** Comparison between ocular adnexa infections and intraocular infection due to *Sporothrix*.

	Ocular Adnexal Sporotrichosis [8,11,12]	Intraocular Sporotrichosis [9]
Epidemiology (most commonly found in)	Eyelid sporotrichosis in China and Peru Conjunctivitis and dacryocystitis in Brazil	The United States and Brazil
Age group	2–89 years	12–75 years
Occurs in children	Yes (frequent)	Yes (rare)
Clinical course	Less aggressive	Aggressive
Site of lesion	Eyelid, conjunctiva, and lacrimal sac	Scleral, cornea, retina, choroid, and optic nervous
Complications	Rare (cases of bulbar conjunctivitis)	Irreversible vision loss (endophthalmitis)
Risk factor	Traumatic inoculation with plant material and contact with cats	Traumatic inoculation with plant material, HIV infection, and residence in a hyperendemic area of sporotrichosis

**Table 2 jof-07-00951-t002:** Most common eyelid lesions findings in a systematic review, case report, and case series of ocular sporotrichosis.

Author [Ref.]/Year	Country	N° of Cases, Sex	Age	Risk Factor	Type Lesion	Clinical Manifestation	Isolated Species
Ramírez-Soto [8]/2016	Peru, Brazil, Japan, Mexico, Costa Rica, and Argentina	53 cases, male (54%)/female (46%)	45 children (78%)/8 adults (22%)	Traumatic inoculation Contact with cats and sporotrichosis cats	Lymphocutaneous (52%) Fixed (32%) Disseminated (1.5%)	Lesions may be papular, nodular, ulcerative, or plaque-like or may show a combination of these features	*Sporothrix schenckii* *Sporothrix* spp.
Zhang et al. [11]/2016	China	72 cases, not specified	43 children (60%)/22 adults (40%)	History of trauma with vegetal material and wood	Lymphocutaneous (57%) Fixed cutaneous (35%) Disseminated (8%)	Lesions may be granuloma annulare-like plaque, abscess, and cyst-like	*S. schenckii*
Fan et al. [28]/2016	China	10 cases, six females	3–81 years (mean 46.5 years)	Not specified	Fixed cutaneous (6 cases) Lymphocutaneous (3 cases) Eyelid abscess (1 case)	Lesions in form nodular, ulcerative, plaque-like, cysts, verrucous lesions, or abscesses	*S. schenckii*
Ramirez-Oliveros et al. [10]/2021	Brazil	Two females	Case 1, 7 years Case 2, 65 years	Sporotrichosis cat in one case	Fixed (2 cases)	One or multiple erythematous papules on the eyelid	*Sporothrix* spp.
Gameiro Filho et al. [26]/2020	Brazil	Female	13 years	Contact with cat	Lymphocutaneous	Eyelid granulomatous conjunctivitis lesion and conjunctivitis	*S. brasiliensis*
Liu et al. [27]/2021	China	Male	50 years	Not specified	Fixed	Small mass on the eyelid with rough keratinization	*S. globosa*
Ahmad-Fauzi et al. [34]/2021	Malaysia	Female	56 years	Exposure to cats	Fixed	Eyelid swelling and erythematous nodules with crusted skin	*S. schenckii*
Wee E [35]/2018	Australian	Female	22 years	Cat scratched her eyelid	Lymphocutaneous	Ulcerated right lower eyelid plaque	*S. schenckii*
Ramírez-Soto et al. [36]/2016	Peru	Male	6 years	Not specified	Lymphocutaneous	Ulcerated right lower eyelid plaque with nodule on the face	*Sporothrix* spp.

**Table 3 jof-07-00951-t003:** Most common findings of conjunctival sporotrichosis in cases report and case series.

Author [Ref], Year	Age/Sex	Country	Clinical Form	Conjunctivitis	Eyelid/Parinaud	Isolated Species	Sequelae
Yamagata et al. [37], 2017	68/F	Brazil	Lymphocutaneous	Granulomatous conjuntivitis (right eye) involving the inferior tarsal and bulbar conjunctiva and conjuntival hyperemia	Parinaud	*Sporothrix* sp.	Fibrosis of the inferior tarsal and bulbar conjunctiva
	46/F	Brazil	Lymphocutaneous	Conjunctival hyperemia with infiltration (left eye)	Periocular edema/Parinaud	*Sporothrix* sp.	Symblefaron on the superior conjunctiva
	14/M	Brazil	Lymphocutaneous	Granulomatous conjunctivitis affecting the lower tarsal conjunctiva (right eye)	Right eyelid edema/Parinaud	*Sporothrix* sp.	--
Madeiros et al. [38], 2016	59/F	Brazil	Disseminated	Pain and burning sensation on the left bulbar conjunctiva	Parinaud	*Sporothrix* spp.	
Hampton et al. [39], 2002	34/M	USA	Lymphocutaneous	1cm diameter, elevated mobile mass without ulceration on the inferotemporal bulbar conjunctiva of the right eye without surrounding episcleral injection	Parinaud	HxPx suppurative granulomatous inflammation and tiny budding yeasts. Culture and electron microscopy (*S. schenckii*)	Asymptomatic and visually insignificant subepithelial corneal infiltrates developed
Schubach et al. [40], 2005	28/F	Brazil	Lymphocutaneous	Granulomatous hyperemic lesion covered with whitish secretion on the lower right conjunctiva, tarsus region	Parinaud	The isolates were on subcultivation	--
	49/F	Brazil	Lymphocutaneous	5 mm granulomatous lesion in the lateral region of the lower right conjunctiva. Conjunctival secretion of 10 days’ duration	Right palpebral edema/Parinaud	Conjunctival swab culture (*S. schenckii*)	
Gameiro et al. [26], 2020	13/F	Brazil	Lymphocutaneous	Follicles in the lower tarsal conjunctiva with conjunctival hyperemia	Nodules in the nasal area of the upper and lower eyelid of the left eye. Lower eyelid swelling/Parinaud	Culture + *S. brasiliensis*	A small scar left the malar region and lower eyelid Margin of the left eye
Lee et al. [41], 2020	15/F	Malaysia	Cutaneous	Multiple nodules with central ulceration over the bulbar and forniceal conjunctiva. Generalized conjunctival injection of the left eye. Ulcerated granulomatous on the upper and lower tarsal conjunctiva	Periorbital edema	PCR + *S schenckii*	
Kashima et al. [42], 2010	62/F	Japan	Extracutaneous	Subconjunctival salmon-pink tumor and conjunctival injection around corneal limbus (left eye)	Palpebral edema	Histopathologically, epithelioid granuloma with microabscesses and infiltration of plasma cells with yeast-like spherules	
Ferreira et al. [43], 2018	78/F	Brazil	Extracutaneous	Infiltration of bulbar conjunctiva but without infiltration of the palpebral conjunctiva	Periorbital edema (left eye)	Culture (conjunctiva) *Sporothrix* spp.	
Reinprayoon et al. [44], 2020	42/F	Thailand	Cutaneous	A chronic, painless ulcerated lesion with a whitish plaque on the left lower tarsal conjunctiva after a keloid-like conjunctival lesion was excised one month ago		PCR (conjunctiva) *S. schenckii.*	
Matos et al. [45], 2020	68/F	Brazil	Lymphocutaneous	A subconjunctival infiltrative lesion in the right eye, occupying the upper and lower fornix	Suppurative nodular lesions on lower right eyelid/Parinaud	*S. brasiliensis.*	Diffuse symblepharon after resolution of the acute phase, with restriction of ocular abduction
Liborio et al. [46], 2020	40/F	Brazil	Lymphocutaneous	Conjunctival granulomas in lower and upper tarsus	Granulomatous lesion on the lower eyelid left eye/Parinaud oculoglandular syndrome		
Arinelli et al. [12], 2019	31.8 ± 23.4/F = 16 (76); M = 5 (24)	Brazil	NS	NS	Eyelids 4 (19); Parinaud 16 (76)	*Sporothrix* sp.	Symblefaron = 3 (14.3); conjuntival fibrosis = 2 (9.5)
Ribeiro et al. [47], 2020	69/F	Brazil	Cutaneous	Conjunctival hyperemia, follicles	Several granulomas in the upper and lower eyelid	Culture of the scrapings and conjunctival secretions + *Sporothrix schenckii*	
	13/F	Brazil	Lymphocutaneous	Conjunctival hyperemia, nodules in the inferior tarsal conjunctiva	Mild upper eyelid edema and granulomas in the lower and upper eyelid (right eye)	Culture of the scrapings and conjunctival secretions + *Sporothrix schenckii*	
	22/M	Brazil	Lymphocutaneous	Granulomatous lesion near the caruncle, mucopurulent secretion and papillae in the tarsal conjunctiva, conjunctival hyperemia	Upper eyelid edema	Culture of the scrapings and conjunctival secretions + *Sporothrix schenckii*	
	18/F	Brazil	Lymphocutaneous	Conjunctival hyperemia 1+/ 4 + serous secretion in the fornix	Mild upper eyelid edema in the right eye; several granulomas in the right lower eyelid, one granuloma in the nasal region of the right upper eyelid. Several granulomas in the temporal area of the right upper eyelid	Culture of the scrapings and conjunctival secretions *Sporothrix schenckii*	
	21/F	Brazil	Extracutaneous	Conjunctival hyperemia (1+/4+), granulomas in the inferior tarsal conjunctiva of the left eye with serous secretion in the fornix		Culture of the scrapings and conjunctival secretions + *Sporothrix schenckii*	
	15/M	Brazil	Lymphocutaneous	A granulomatous lesion in the inferior tarsal conjunctiva, and papillae in the tarsal, conjunctiva with serous secretion in the fornix; conjunctival hyperemia		Culture of the scrapings and conjunctival secretions + *Sporothrix schenckii*	
Freitas et al. [48], 2012	59/M	Brazil	Disseminated	Granulomatous conjunctivitis of the right eye		Skin exudate conjunctival swab	
Lemes et al. [49], 2020	3/M	Brazil	Lymphocutaneous	Granulomatous erythematous lesion on the lower portion of the tarsal conjunctiva	Parinaud ocular syndrome	Culture + *Sporothrix* spp.	
	12/M	Brazil	Lymphocutaneous	Conjunctivitis on the left eye (sneezing by a sick cat)		Culture + *Sporothrix* spp.	
Ling et al. [50], 2018	18/F	Malaysia	Lymphocutaneous	Granulomatous conjunctival lesion covered with thin whitish discharge	Several small nodular lesions on the left inferior palpebral conjunctiva	Culture (conjunctival fornix) + *S. schenckii.*	
Ferreira et al. [51], 2014	21/M	Brazil	Disseminated	A lesion in the lower tarsal conjunctiva	Parinaud’s oculoglandular syndrome	Culture of hand lesion, *Sporothrix schenckii* (+)	
Ribeiro et al. [52], 2010	34/F	Brazil	Lymphocutaneous	Nodules in inferior tarsal conjunctiva	Parinaud’s oculoglandular syndrome	Culture of hand lesion, *Sporothrix schenckii* (+)	
Alvarez and Lopez-Villegas [53], 1964	11/M	Colombia	Extracutaneous	Temporal bulbar conjunctiva	Temporal bulbar conjunctiva	Culture of hand lesion, *Sporothrix schenckii* (+)	
Ahmad et al. [34], 2021	36.5 ± 19.2/F = 4 (67); M = 2 (33)	Malaysia	NS	GC on palpebral conjunctiva	Right 3 (50), Left 3 (50). Eyelids 2 (33)	Culture + *Sporothrix* sp.	Symblefaron = 2 (33); skin scarring = 1 (16.6)
Paiva et al. [54], 2020	25/F	Brazil	Lymphocutaneous	Upper eyelid edema and nodular lesions, GC, and fistulizing dacryocystitis	Palpebral nodules. The left bulbar and lower tarsal conjunctiva	Culture + *Sporothrix* sp.	Persistent fistula
Lacerda et al. [55], 2019	25/M	Brazil	Lymphocutaneous	Acute redness with conjunctival granulomatous lesions	Left tarsal conjunctiva	PCR (conjunctiva) *S. brasiliensis*.	

F: female; M: male; SD: standard deviation; NS: not specified; GC: granulomatous conjunctivitis.

**Table 4 jof-07-00951-t004:** Consolidated of the clinical-epidemiological findings of conjunctival sporotrichosis.

	Tarsal Conjunctiva [26,34,37,40,44,47,49,50,51,52]	Bulbar Conjunctiva [38,39,43,53]
Age group	3–69 years	11–78 years
Gender	Female (63%)	Male:Female 1:1
Occurs in children	Yes (frequent) 36.8%	Yes (rare) 25%
Clinical course	Less aggressive	More aggressive
Type of lesion (most typical)	Lymphocutaneous or fixed	Lymphocutaneous, fixed, or disseminated
Primary ocular infection	Frequent: tarsal/bulbar conjunctivitis Rare: tarsal conjunctivitis alone	Rare: bulbar conjunctivitis alone
Complications	36.8% of cases	25% of cases
Treatment duration with itraconazole (months)	100 mg = 9.5; 200 mg = 5.5; 400 mg = 5.5	100 mg =1; 200 mg = 2; 300 mg = 6; potassium iodide = 3
Cure (%)	79	75

**Table 5 jof-07-00951-t005:** Comparison between the exogenous and endogenous endophthalmitis due to *Sporothrix* [9].

	Exogenous Endophthalmitis	Endogenous Endophthalmitis
Epidemiology	United States	Brazil and the United State
Pathogen	*S. schenckii*	*S. schenckii* and *S. brasiliensis*
Risk factor	Traumatic inoculation	HIV-infection and residence in a hyperendemic area
Age group	13–75 years	12–56 years
Occurs in children	Infrequent	Infrequent
Disease duration at diagnosis	123.6 days	76.5 days
Ocular manifestation	Scleritis, anterior and posterior uveitis	Anterior and posterior uveitis, panuveitis, choroiditis, choroidoretinitis, and retinitis
Clinical course	Aggressive	Aggressive
Complications	Blindness and irreversible vision loss	Blindness and irreversible vision loss

**Table 6 jof-07-00951-t006:** Primary differential diagnoses of ocular sporotrichosis.

Type of Disease, Agent	Ocular Adnexal Sporotrichosis	Intraocular Sporotrichosis
Fungal infection	Palpebral mycoses, including paracoccidioidomycosis, cutaneous blastomycosis, cutaneous coccidioidomycosis, cutaneous histoplasmosis, and cryptococcosis. Fungal conjunctivitis, including aspergillosis, blastomycosis, candidiasis, coccidioidomycosis, and cryptococcosis.	Exogenous endophthalmitis, including infection by *C. albicans*, *C. glabrata*, *C. parapsilosis*, *Aspergillus fumigatus*, *A. terreus*, *Fusarium oxysporum*, *F. solani*, *Acremonium strictum*, and *Phialophora verrucosa*. Endogenous endophthalmitis, including infection by yeasts (*C. albicans*, *C. parapsilosis*, and *C. glabrata*), dimorphic molds (*Blastomyces dermatitidis*, *Histoplasma capsulatum*, and *Coccidioides immitis*), dematiaceous molds (*S. apiopsermum*, *S. prolificans*, *Cladophialophora bantiana*, and *Phialemoniunm curvatum*).
Bacterial infection	Bacterial conjunctivitis by *Haemophilus influenzae*, *Streptococcus pneumoniae*, and *Moraxella catarrhalis*. Bacterial dacryocystitis.	Exogenous endophthalmitis including infection by *Bacillus cereus*, coagulase-negative staphylococci. Endogenous endophthalmitis including infection by *Staphylococcus aureus*, streptococci, Gram-negative bacilli such as *Klebsiella*, syphilis, and tuberculosis.
Viral infection	Viral conjunctivitis due to Herpes simplex and zoster virus.	Retinitis, choroiditis, and uveitis caused by cytomegalovirus, herpes simplex, and herpes zoster.
Non-infectious	Palpebral lesions include chalazions, hordeolums, and malignant tumors. Conjunctivitis include Parinaud’s oculoglandular syndrome. Lacrimal sac tumor.	Pseudoendophthalmitis from intravitreal injections, uveal melanoma, retinoblastoma, and sarcoidosis.

## Data Availability

Not applicable.

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
