# Peer review of "Ocular Sporotrichosis"

_jof, 2021, doi:10.3390/jof7110951_

Round 1

Reviewer 1 Report

In this work, the authors show a review of data on ocular sporotrichosis. In the last two decades, the geographic expansion and the increase in the number of cases of sporotrichosis make rare extracutaneous forms become more frequent. A challenge for doctors then arises: suspect and identify cases of ocular sporotrichosis, and then treat these patients properly avoiding further sequelae. Thus, I consider the work relevant, but a major review must be carried out.

-The legend of figure 1 must be improved for a better understanding of the figure. Mainly, in the representation of cases of ocular sporotrichosis. I recommend including the values used in this representation (percentage, absolute values?)

-Lines 89-90 – english revision

-Lines 93-94 – Although the sentence describes the clinical presentation, there is a misquotation of figure 1 (global distribution map).

- The text, table 1 and figure 2 need English revision.

- Include in Figure 2 that there are other extracutaneous manifestations of sporotrichosis in addition to ocular sporotrichosis.

- How were the percentage values calculated in Figure 2? Its meaning was not clear.

- The titles in tables 3 and 4 must be reorganized. For example: Table 3 - consolidated of the clinical-epidemiological findings of conjunctival sporotrichosis. Table 4 - most common findings of conjunctival sporotrichosis in a systematic review, case report and case series of ocular sporotrichosis.

-Line 325 - replace “table 5” with “table 6”

- If possible, include some pictures of ocular sporotrichosis.

- Some references on ocular sporotrichosis were not included in this review. They are listed below:

Human sporotrichosis outbreak caused by Sporothrix brasiliensis in a veterinary hospital in Southern Brazil.

Xavier JRB, Waller SB, Osório LDG, Vives PS, Albano APN, Aguiar ESV, Ferreira MRA, Conceição FRD, Faria RO, Meireles MCA, Gomes ADR.J Mycol Med. 2021 Sep;31(3):101163. doi: 10.1016/j.mycmed.2021.101163. Epub 2021 Jun 2.PMID: 34157511

Different clinical manifestations of ocular sporotrichosis in the same patient: an alert to ophthalmologists in nonendemic areas.

Paiva ACM, Biancardi AL, Curi ALL.Arq Bras Oftalmol. 2020 Sep-Oct;83(5):457-458. doi: 10.5935/0004-2749.20200107.

High-Virulence Cat-Transmitted Ocular Sporotrichosis.

Lacerda Filho AM, Cavalcante CM, Da Silva AB, Inácio CP, de Lima-Neto RG, de Andrade MCL, Magalhães OMC, Dos Santos FAG, Neves RP.Mycopathologia. 2019 Aug;184(4):547-549. doi: 10.1007/s11046-019-00347-6. Epub 2019 Jun 22.PMID: 31230198

Author Response

We thank the Reviewer for your comments and constructive criticism, we believe that the quality of our manuscript has been significantly improved. We have revised our paper in a point-by-point manner. Modifications are in yellow text.

Reviewer #1: In this work, the authors show a review of data on ocular sporotrichosis. In the last two decades, the geographic expansion and the increase in the number of cases of sporotrichosis make rare extracutaneous forms become more frequent. A challenge for doctors then arises: suspect and identify cases of ocular sporotrichosis, and then treat these patients properly avoiding further sequelae. Thus, I consider the work relevant, but a major review must be carried out.

Comment 1: The legend of figure 1 must be improved for a better understanding of the figure. Mainly, in the representation of cases of ocular sporotrichosis. I recommend including the values used in this representation (percentage, absolute values?)

Response 1: Thank you for your comment. We have included your suggestions “absolute values (n = N° cases)”

Comment 2: Lines 89-90 – english revision

Response 2: Thank you for your comment. We have corrected this error.

Comment 3: Lines 93-94 – Although the sentence describes the clinical presentation, there is a misquotation of figure 1 (global distribution map).

Response 3: Thank you for your comment. We have corrected this error.

Comment 4: The text, table 1 and figure 2 need English revision.

Response 4: Thank you for your comment. We have corrected the errors in Table 1 and Figure 2.

 Comment 5: Include in Figure 2 that there are other extracutaneous manifestations of sporotrichosis in addition to ocular sporotrichosis.

Response 5: Thank you for your comment. We have included your suggestions (see Figure 2)

Comment 6: How were the percentage values calculated in Figure 2? Its meaning was not clear.

Response 6: Thank you for your comment. We have included your suggestions. We remove the values. We only describe the classification of sporotrichosis (see Figure 2)

 Comment 7: The titles in tables 3 and 4 must be reorganized. For example: Table 3 - consolidated of the clinical-epidemiological findings of conjunctival sporotrichosis. Table 4 - most common findings of conjunctival sporotrichosis in a systematic review, case report and case series of ocular sporotrichosis.

Response 7: Thank you for your comment. We have included your suggestions (see Tables 3 and 4)

 Comment 8: Line 325 - replace “table 5” with “table 6”

Response 8: Thank you for your comment. We have corrected the error.

 Comment 9: If possible, include some pictures of ocular sporotrichosis.

Response 9: Thank you for your comment. We have included your suggestions. We include a panel of figures (Figure 3A-H).

Comment 10: Some references on ocular sporotrichosis were not included in this review. They are listed below:

Xavier JRB, Waller SB, Osório LDG, Vives PS, Albano APN, Aguiar ESV, Ferreira MRA, Conceição FRD, Faria RO, Meireles MCA, Gomes ADR. Human sporotrichosis outbreak caused by Sporothrix brasiliensis in a veterinary hospital in Southern Brazil. J Mycol Med. 2021 Sep;31(3):101163. doi: 10.1016/j.mycmed.2021.101163. Epub 2021 Jun 2.PMID: 34157511

Paiva ACM, Biancardi AL, Curi ALL. Different clinical manifestations of ocular sporotrichosis in the same patient: an alert to ophthalmologists in nonendemic areas. Arq Bras Oftalmol. 2020 Sep-Oct;83(5):457-458. doi: 10.5935/0004-2749.20200107.

Lacerda Filho AM, Cavalcante CM, Da Silva AB, Inácio CP, de Lima-Neto RG, de Andrade MCL, Magalhães OMC, Dos Santos FAG, Neves RP. High-Virulence Cat-Transmitted Ocular Sporotrichosis. Mycopathologia. 2019 Aug;184(4):547-549. doi: 10.1007/s11046-019-00347-6. Epub 2019 Jun 22.PMID: 31230198

Response 10: Thank you for your comment. We have included two references (Paiva and Lacerda). Xavier et al. does not describe the epidemiological clinical characteristics of the patients.

Reviewer 2 Report

The authors present a classic review describing the clinical aspects, diagnosis and treatment of ophthalmologic lesions of sporotrichosis, that have been increasingly found in clinical practice in the transmission areas. The manuscript draws attention to sporotrichosis in the ocular region and its adnexa, and may be of interest to general clinicians and ophthalmologists in endemic areas. My only suggestion is include an explanation about the eyebrow as ocular adnexa, since eyebrows are not always considered as part of ocular adnexa. I have asked ophthalmologists and the majority does not consider eyebrows as part of their specialty. As the group has already published similar data (Am. J. Ophtalm., 2015) and in this article maintains the eyebrows as part of the ocular adnexa (abstract, page 3-lines 91-93, Page 4-lines 109-111, and in the results where the percentage of lesions on the eyebrows is presented), It could be clarified by adding add an explanatory sentence in item 2.1.

Author Response

We thank the Reviewer for your comments and constructive criticism, we believe that the quality of our manuscript has been significantly improved. We have revised our paper in a point-by-point manner. Modifications are in yellow text.

Reviewer #2: The authors present a classic review describing the clinical aspects, diagnosis and treatment of ophthalmologic lesions of sporotrichosis, that have been increasingly found in clinical practice in the transmission areas. The manuscript draws attention to sporotrichosis in the ocular region and its adnexa, and may be of interest to general clinicians and ophthalmologists in endemic areas.

Comment 1: My only suggestion is include an explanation about the eyebrow as ocular adnexa, since eyebrows are not always considered as part of ocular adnexa. I have asked ophthalmologists and the majority does not consider eyebrows as part of their specialty. As the group has already published similar data (Am. J. Ophtalm., 2015) and in this article maintains the eyebrows as part of the ocular adnexa (abstract, page 3-lines 91-93, Page 4-lines 109-111, and in the results where the percentage of lesions on the eyebrows is presented), It could be clarified by adding add an explanatory sentence in item 2.1.

Response 1: Thank you for your comment. We have included your suggestions. We have eliminated the term eyebrows.

Reviewer 3 Report

The manuscript #jof-1435684, entitled “Ocular Sporotrichosis” by Ramírez-Soto et al. presents a comprehensive review on (in line with the title) rarely, but emerging cases of ocular sporotrichosis. The topic is of a great importance as ocular infection due to Sporothrix may lead to a sight loss.

The manuscript is logically planned and consist of introducing Sporothrix fungi, explaining different types of ocular Sporothrix infections, as well as presenting current diagnosis and treatment options. The quality of presentation and overall scientific soundness are on a high level. Most of my comments concern minor (mostly editorial) issues.

Line 50 and 52: "Brasil" and "Brazil" should be unified.

Line 51: The authors stated that "Sporotrichosis has been described both in humans and animals like cats and dogs". Does animal sporotrichosis occur in all mentioned areas or it is a worldwide phenomena.

Line 101: Missing dot in Figure caption.

Line 104: Missing dot in Figure caption.

Line 136: Missing dot in Table caption.

Line 176: "dacryocystitis" should start with capitalized letter. 

Line 239: Missing dot in Table caption.

Line 241: Delete dot in the chapter title.

Line 253: "Sporothirx" should be italicized.

Line 269: 107 instead of "5 × 107 cells".

Line 298-302: The authors stated that the usage of amphotericin is "the treatment of choice". Later in the text the authors mention that "Treatment time is variable and depends more on side effects (kidney)". In my opinion the authors should elaborate more on possible side effects of amb treatment on human tissues and thus refer to its status as a treatment of choice.

Author Response

We thank the Reviewer for their comments and constructive criticism, we believe that the quality of our manuscript has been significantly improved. We have revised our paper in a point-by-point manner. Modifications are in yellow text.

Reviewer #3: The manuscript #jof-1435684, entitled “Ocular Sporotrichosis” by Ramírez-Soto et al. presents a comprehensive review on (in line with the title) rarely, but emerging cases of ocular sporotrichosis. The topic is of a great importance as ocular infection due to Sporothrix may lead to a sight loss. The manuscript is logically planned and consist of introducing Sporothrix fungi, explaining different types of ocular Sporothrix infections, as well as presenting current diagnosis and treatment options. The quality of presentation and overall scientific soundness are on a high level. Most of my comments concern minor (mostly editorial) issues.

Comment 1: Line 50 and 52: "Brasil" and "Brazil" should be unified.

Response 1: Thank you for your comment. Thank you for your comment. We have corrected the error.

Comment 2: Line 51: The authors stated that "Sporotrichosis has been described both in humans and animals like cats and dogs". Does animal sporotrichosis occur in all mentioned areas or it is a worldwide phenomena.

Response 2: Thank you for your comment. We have included two references (lines 49-51)

Comment 3: Line 101: Missing dot in Figure caption.         

Response 3: Thank you for your comment. We have included your suggestions.

Comment 4: Line 104: Missing dot in Figure caption.

Response 4: Thank you for your comment. We have included your suggestions.

Comment 5: Line 136: Missing dot in Table caption.

Response 5: Thank you for your comment. We have included your suggestions

Comment 6: Line 176: "dacryocystitis" should start with capitalized letter. 

Response 6: Thank you for your comment. We have included your suggestions

Comment 7: Line 239: Missing dot in Table caption.

Response 7: Thank you for your comment. We have included your suggestions

Comment 8: Line 241: Delete dot in the chapter title.                

Response 8: Thank you for your comment. We have included your suggestions.

Comment 9: Line 253: "Sporothirx" should be italicized.

Response 9: Thank you for your comment. We have included your suggestions.

Comment 10: Line 269: 107 instead of "5 × 107 cells".

Response 10: Thank you for your comment. We have included your suggestions (line 288).

Comment 11: Line 298-302: The authors stated that the usage of amphotericin is "the treatment of choice". Later in the text the authors mention that "Treatment time is variable and depends more on side effects (kidney)". In my opinion the authors should elaborate more on possible side effects of amb treatment on human tissues and thus refer to its status as a treatment of choice.

Response 11: Thank you for your comment. We have included your suggestions (lines 324-327)

Round 2

Reviewer 1 Report

Given the modifications made by the authors, I consider the work suitable for publication.